# Phylogenetic and Codon Usage Bias Analysis Based on mt-DNA of *Cyphochilus crataceus* (Coleoptera: Melolonthinae) and Its Neighboring Species

**DOI:** 10.3390/genes16020111

**Published:** 2025-01-21

**Authors:** Haofeng Zhan, Quan Cao, Xiaofei Yang

**Affiliations:** 1College of Tea Science, Guizhou University, Guiyang 550025, China; zhanchenhan@126.com (H.Z.); wyquancao@126.com (Q.C.); 2Jiangsu Key Laboratory of Sericutural Biology and Biotechnology, School of Biotechnology, Jiangsu University of Science and Technology, Zhenjiang 212000, China

**Keywords:** *C. crataceus*, mitochondrial genome, phylogenetic analysis, codon usage bias analysis

## Abstract

**Background/Objectives**: In order to determine the basic structural characteristics of the mitochondrial genome of *Cyphochilus crataceus* and explore its phylogenetic status, as well as to understand the codon usage bias of Melolonthinae species, the next-generation sequencing was used to obtain the mitochondrial genome sequence of *C. crataceus*. **Methods**: Combined with 121 sequences of Scarabaeidae downloaded from GeneBank, a phylogenetic tree of the family was constructed using PhyloSuite v 1.2.3 software. Additionally, the codon composition and codon usage bias of the mitochondrial protein-coding genes of *C. crataceus* and 16 other Melolonthinae species were analyzed. **Results**: The results showed that the mitochondrial genome sequence of *C. crataceus* was 17,946 bp in length, with an A + T content of 71.82%, exhibiting a significant AT bias and a preference for ending with the base A/U, showed typical features of Scarabaeidae mitogenomes. The analysis of RSCU, ENC-plot, and neutrality plot revealed that factors such as nucleotide composition, gene mutations, and natural selection can have an impact on codon usage bias, but the intensity varies. For *C. crataceus*, codon usage preference is primarily influenced by gene mutations. The phylogenetic tree results indicated that, apart from Melolonthinae, all other subfamilies within Scarabaeidae were monophyletic. **Conclusions**: This study not only enriches the mitochondrial genome information of scarab beetles in the subfamily Melolonthinae but also provides important foundational information for molecular systematics, population genetics, and molecular ecology research in the family Scarabaeidae.

## 1. Introduction

*C. crataceus* (Niijima & Kinoshita, 1923) [1], belongs to Melolonthinae, Scarabaeidae, and Coleoptera. Scarabaeidae is a large group of insects within Coleoptera. Based on their feeding habits, beetles can be divided into two branches: coprophagous and phytophagous (including saprophagous). *C. crataceus* is a phytophagous beetle reported to damage the leaves and roots of oil tea plants. It is mainly distributed in the mid to low-altitude mountainous regions south of China and bears a striking resemblance to *Cyphochilus taipeiensis*. Research on *C. crataceus* has mainly focused on synthesizing high-performance materials by mimicking the iridescent scales of the beetle, while its distribution and harmful effects remain unclear. Since Erichson [2], phytophagous beetles have been named Pleurosticti and are considered an independent branch, receiving affirmation from morphological studies, including Melolonthinae, Cetoniinae, Dynastinae, and Rutelinae as the four subfamilies [3,4]. However, the taxonomy and phylogeny of Melolonthinae are still controversial. Based on morphological characteristics, multi-gene sequences, and mitochondrial genome analysis, Melolonthinae is supported as a monophyletic group [5,6,7]. Nevertheless, some authors have proposed merging Melolonthinae, Rutelinae, and Dynastinae into the family Melolonthidae [8,9]. There is also debate on whether Sericinae should be considered as a tribe within Melolonthinae [5,6,10]. Clarifying the status of subgroups within the family Melolonthinae may help understand their evolution and aid in future classification work; thus, it is necessary to integrate existing data for a more comprehensive reconstruction of the phylogenetic tree of Scarabaeidae.

Mitogenome is a useful tool for resolving phylogenetic issues at different taxonomic levels in animals and plants [11,12]. The complete mitochondrial genome of Melolonthinae is relatively conservative, with most showing no rearrangements or deletions, including 13 protein-coding genes (PCGs), 22 transfer RNA genes (tRNAs), 2 ribosomal RNA genes (rRNAs), and 1 control region. Next-generation sequencing (NGS) technologies have been widely used to reconstruct mitochondrial genomes of various insects, including Scarabaeidae [7,13,14,15]. Organisms with close relationships tend to exhibit similar codon usage patterns, and codon usage bias analysis can reveal the origin of species or genes, mutation patterns, and phylogenetic relationships among organisms [16]. Genome composition is one of the factors influencing codon usage bias [17]. In this study, we assembled the mitochondrial genome of *C. crataceus* using NGS technology. After assembly and annotation, we analyzed the structure and gene arrangement, composition, codon usage bias, and influencing factors of the mitochondrial genome of *C. crataceus*, as well as the structure of transfer RNA (tRNA). By combining published data on Scarabaeidae mitochondrial genomes, we reexamined the phylogenetic relationships of Scarabaeidae, with a particular focus on the monophyly of Melolonthinae and the phylogenetic position of Sericinae. Through the analysis of mitochondrial genome composition, codon usage bias, and influencing factors, we aim to delve deeper into elucidating the phylogenetic position of *C. crataceus* and clarifying the taxonomic status of Melolonthinae.

## 2. Materials and Methods

### 2.1. Sample Collection and Mitogenome Sequencing, Assembly, and Annotation

A small leg muscle tissue sample was extracted from *C. crataceus* collected from the Yinggeling National Nature Reserve, Hainan Province, China (18°49′40.49″ N, 109°31′49.89″ E). The remaining sample was stored in 75% alcohol and kept refrigerated in a −20 °C freezer at the College of Tea Science of Guizhou University. Genomic DNA was extracted using the standardized CTAB method [18] and then subjected to next-generation sequencing by Shanghai Personal Biotechnology Co., Ltd. (Shanghai, China) using an Illumina NovaSeq sequencing platform (Illumina NovaSeq, Illumina Inc., San Diego, CA, USA) with 300 bp paired ends. The final mitochondrial sequence was corrected using pilon v.1.18 software [19], assembled using A5-miseq version 2.0 [20], annotated using Geneious Prime 2020.2.4 [21], and deposited into GenBank under the accession number OP963801.1 The CGview Server http://stothard.afns.ualberta.ca/cgview_server/index.html (accessed on 25 September 2017)was utilized to generate the circular map of the mitochondrial genome [22].

### 2.2. Composition, Structure, and Codon Usage Bias Analysis of Mitochondrial Genomes

The MITOS web server http://mitos2.bioinf.uni-leipzig.de/index.py (accessed on 25 September 2017) [23] and tRNAscan-SE v1.3.1 [24] were utilized for predicting the secondary structures of tRNAs. Base compositions, codon usage, and relative synonymous codon usage (RSCU) were analyzed using MEGA v.7.0 [25] and graphically represented using ggplot2 packaging in R. The strand asymmetry was calculated using the formulas: AT-skew = (A − T)/(A + T) and GC-skew = (G − C)/(G + C) [26]. Factors influencing codon usage bias were analyzed using ENC-plot and neutrality plot analysis. The formula for calculating the expected ENc value is: ENC = 2 + GC3s + 29/[GC3s2 + (1 − GC3s)2]. A lower ENC value indicates a stronger influence of GC3s on codon usage bias, while a higher ENC value suggests a stronger impact of natural selection on gene evolution [27]. The neutrality plot plots GC12 (average of GC1 and GC2) against GC3, and the correlation between GC3 and GC12 reflects the influence of base mutations on codon usage bias. A stronger correlation indicates a greater impact of base mutations, while a weaker correlation suggests a higher influence of selection pressure. The analysis was performed using MEGA v.7.0 [25] to calculate the GC1, GC2, and GC3 values for each codon.

### 2.3. The Phylogenetic Tree of Scarabaeidae

To construct the phylogenetic tree of Scarabaeidae, 122 complete mitochondrial genome sequences of Scarabaeidae were downloaded from the NCBI database. The sequence of *C. crataceus* obtained in this study was also included, with the insect *Hydrochus carinatus* from the family Hydrochidae used as an outgroup. Detailed information on the data is provided in Appendix A. The phylogenetic tree was constructed using the PhyloSuite v 1.2.3 software (downloaded from https://github.com/dongzhang0725/PhyloSuite/releases) [28,29]. Initially, the 13 protein-coding genes (PCGs) sequences were extracted and aligned using the MAFFT program (version 7.313) [30] included in PhyloSuite software. The alignment results were concatenated into a combined sequence dataset. PartitionFinder2 [31] was then used to precisely calculate the partitioning strategy and corresponding evolutionary models. The branch lengths were set to converge, and the Bayesian model and Akaike information criterion (AICc) were used to select the optimal solution using a greedy search strategy. The phylogenetic tree was constructed using both Bayesian inference (BI) and Maximum Likelihood (ML) methods. In the BI analysis, the Bayesian inference method was used to construct the phylogenetic tree with two parallel runs and four chains. After running for 1,000,000 generations, the average standard deviation of split frequencies (ASDOSFs) values was 0.005 (<0.01). Sampling was performed every 1000 generations, and 25% of the initial samples were discarded as burn-in [32]. Posterior probability (PP) values were calculated for each node. The ML tree was generated in IQ-tree [33] mode using the PhyloSuite program. Ultrafast bootstrapping with 10,000 replicates was performed using the Edge-linked partition style [34,35] to calculate bootstrap (BS) values for each node. The final phylogenetic trees were visualized and refined in Adobe Illustrator CS5.

## 3. Results

### 3.1. Analysis of Mitochondrial Genome Composition and Features

The complete mitochondrial genome of *C. crataceus* is 17,946 bp in length (Figure 1), containing 13 protein-coding genes (PCGs), 1 D-loop region, 22 tRNAs, and 2 rRNAs. The sequence did not exhibit any rearrangement and showed typical features of Scarabaeidae mitogenomes. Among them, *trnQ*, *trnC*, *trnY*, *trnF*, *nad5*, *trnH*, *nad4*, *nad4l*, *trnP*, *nad1*, *trnL1*, *rrnL*, *trnV*, and *rrnS* are located on the J strand, while *trnI*, *trnM*, *nad2*, *trnW*, *cox1*, *trnL2*, *cox2*, *trnK*, *trnD*, *atp8*, *atp6*, *cox3*, *trnG*, *nad3*, *trnA*, *trnR*, *trnN*, *trnS1*, *trnE*, *trnT*, *nad6*, *cob*, *trnS2*, and *OH* are located on the N strand (Table 1). Like other insect mitochondrial genomes, the *C. crataceus* mitochondrial genome also exhibits gene overlaps and intergenic regions, with 19 instances of base overlaps ranging from 1–8 bp and 11 instances of base gaps ranging from 1–1531 bp (Table 1). The overall genome composition includes A: 37.64%, C: 18.3%, G: 9.88%, T: 34.19%, (A + T): 71.82%, (G + C): 28.18%, with an AT skew (0.048) greater than the GC skew (−0.299), indicating a significant AT bias. However, certain sequences such as *nad1*, *nad4*, *nad4L*, *nad5*, and *rRNA* exhibit significant GC bias, while some genes show both AT skew < 0 and GC skew < 0 (Table 2). The lengths of the 13 PCGs in *C. crataceus* range from 156 bp to 1718 bp. Details of the start and stop codons of PCGs in *C. crataceus* are provided in Table 1. The predominant start codons used were ATT for *nd2*, *cox1*, and *nad5*, ATG for *cox2*, *atp6*, *cox3*, *nad4*, *nad4l*, *cytb*, and *nad1*, ATC for *atp8*, and ATA for *nad3* and *nad6*. The commonly used stop codons were TAA for *cox1*, *cox2*, *atp8*, *cox3*, *nad5*, *nad4*, *nad4l*, *atp6*, *nad6*, and *nad1*, whereas *nad2*, *atp6*, *nad3*, and *cytb* ended with TAG. The relative synonymous codon usage (RSCU) value of *C. crataceus* is highly similar to other Melolonthinae species (Figure 2), with the most frequently used codons being GCU, CGA, GGA, UUA, CCU, AGA, UCU, GUA, and GUU (Figure 2). The lengths of the 22 tRNAs range from 63 bp to 71 bp (Table 1), with most *tRNAs* of *C. crataceus* exhibiting the canonical cloverleaf structure, for *trnS1*, which has an unpaired stretch of 10 nucleotides instead of the DHU arm (Figure 3).

### 3.2. Analysis of Factors Influencing Codon Usage Bias

The ENC-GC3 plot was utilized to analyze the evolutionary pressure experienced by Melolonthinae species. Scatter plots were used to depict the relationship between the Expected Number of Codons (ENCs) and the GC content at the third codon position (GC3s) for each species (Figure 4). The ENC-GC3 plots of 17 Melolonthinae species varied, with most protein-coding genes having ENC values lower than the expected values, ranging between 25% and 62%, and GC3 content ranging from 5% to 50%. Some genes exhibited a distribution of codons along the expected curve, indicating that these genes were primarily influenced by base mutations. Points located further from the curve on either side indicated a stronger influence of selection, with a few genes positioned significantly below the curve, suggesting that nucleotide composition, gene mutations, and natural selection, among other factors, influenced codon selection in the 17 Melolonthinae species. In the sequenced *C. crataceus* sequence, genes such as *cox1*, *cox2*, *cox3*, and *atp6* were mainly influenced by base mutations, while *nad1*, *nad2*, *nad3*, *nad4*, *nad4L*, and *nad5* were positioned below the curve, far from the expected ENC value, indicating that the codon usage bias of these genes was influenced by factors such as natural selection.

Neutral plot analysis, reflecting the mutational selection balance of codon usage in Melolonthinae, was conducted based on the correlation between GC12 and GC3 (Figure 4). Analysis of the neutral plots of GC12 and GC3 in Melolonthinae species (Figure 5) showed varying slopes of the standard curves for all 17 species, indicating differences in the intensity of mutational pressure or natural selection influencing codon usage bias. The correlation between GC12 and GC3 content varied among species, with some showing positive correlations and others negative correlations, although the correlations were not significant, further indicating differences in the composition of the third base of codons compared with the first and second bases. The slope of the standard curve for *C. crataceus* in this sequencing was 1.075, close to 1, indicating no difference in the base composition at the three positions of codons, with mutations being the primary force influencing codon bias.

### 3.3. Phylogenetic Analyses of Scarabaeidae

Using the 13 protein-coding genes (PCGs), phylogenetic trees of 122 species of Scarabaeidae were constructed using the RAxML and MrBayes methods, resulting in identical topologies within the major clades (Figure 6). Apart from the outgroups, the tree is primarily divided into two major clades. One clade consisted of the Scarabaeinae and Aphodiinae, both comprising coprophagous beetles, while the other clade included ((((Dynastinae + Rutelinae) + Cetoniinae) + Melolonthinae) + Sericinae), all being phytophagous beetles. Cetoniinae, Dynastinae, Rutelinae, and Sericinae were found to be monophyletic with strong support values (PP: 1, BS: 100). In contrast, Melolonthinae was paraphyletic, consisting of two distinct clades. Within the Melolonthinae branch, *C. crataceus* was most closely related to *Miridiba trichophora* and *Rhopaea magnicornis*, with *C. crataceus* forming a sister clade with these two species, followed by a sister group relationship with *Melolontha hippocastani*, *Polyphylla mandshurica*, *Polyphylla gracilicornis*, and *Polyphylla laticollis*, forming the phylogenetic structure of ((((*P. gracilicornis* + *P. laticollis*) + *P. mandshurica*) + *M. hippocastani*) + *C. crataceus*) + *M. trichophora*.

## 4. Discussion

Previous studies have extensively reported the phylogenetic relationships within Scarabaeidae. However, with the continual addition of new mitochondrial genome sequences within this family, there is a need for a more comprehensive understanding of the accurate phylogenetic positions of the various subfamilies within Scarabaeidae. In this study, we sequenced the mitochondrial genome of *C. crataceus*, a species within the Melolonthinae subfamily that has been underexplored and reported less frequently. By integrating all publicly available accurate mitochondrial genome sequences of Scarabaeidae, we collectively constructed a phylogenetic tree based on the 13 protein-coding genes. Additionally, we conducted a comprehensive analysis of the mitochondrial genome composition, structure, codon usage bias, and influencing factors of *C. crataceus* to further understand its phylogenetic position within the Melolonthinae subfamily and confirm the factors influencing codon usage bias. The arrangement of the *C. crataceus* mitochondrial genome and the secondary structures of the *rrnL* and *rrnS* genes were found to be similar to other Scarabaeidae species [7]. The protein-coding genes of *C. crataceus* exhibited an AT content of 71.82%, showing a clear AT bias with a preference for codons ending in A/U. This study’s results were consistent with those reported by Long et al. [15]. for species such as *Sophrops peronosporus* Gu and Zhang, *Anomala rufventris* Kollar and Redtenbacher, and *Callistethus plagiicollis* Fairmaire, all of which had an AT content exceeding 70% and a preference for A/U-ending codons. Therefore, the codon usage bias in *C. crataceus* and Melolonthinae species is closely related to nucleotide composition, serving as an important factor influencing codon usage bias. The ENC-GC3 plot analysis revealed that the codon usage bias in Melolonthinae species is influenced by multiple factors such as nucleotide composition, gene mutations, and natural selection. The neutral plot analysis indicated that the codon usage bias in Melolonthinae is influenced by mutational pressure or natural selection, with varying intensities, which may be related to the broad distribution range and survival environments of the species.

The phylogenetic analyses of the 122 Scarabaeidae species were largely consistent with previous studies based on rRNAs, nuclear genes, and the 13 PCGs sequences [15,35], phylogenetic structure of ((Scarabaeinae + Aphodiinae) + ((((Dynastinae + Rutelinae) + Cetoniinae) + Melolonthinae) + Sericinae)) + outgroup. This obtained topological structure is consistent with the phylogenetic structures based on mitochondrial nucleotide sequences and amino acid sequences, based on transcriptomic data, and based on the combination of mitochondrial and nuclear genes [7,36,37,38,39,40], indicating that using only mitochondrial protein-coding genes is a simple and effective method for constructing the phylogeny of Scarabaeidae. The use of the Hydrochidae insect *H. carinatus* as an outgroup did not lead to long branch attraction issues, indicating the rational selection of the outgroup species. Melolonthinae was found to be paraphyletic, consistent with results based on morphological features [5], while the other subfamilies were monophyletic, consistent with the findings of most researchers [7,15,36]. However, some studies have suggested that Rutelinae may be a paraphyletic group [7,40], with a possibility of some Dynastinae species being reclassified under Rutelinae [3]. This divergence in results may be due to the limited sample size leading to misinterpretation of the results. Therefore, we do not currently support the proposal to merge Melolonthinae, Rutelinae, and Dynastinae into the family Melolonthidae, as suggested by Cherman et al. [9] The phylogenetic tree in this study did not reveal any species closely related to the sequenced *C. crataceus*, highlighting the importance of sampling more Scarabaeidae species, especially the Aphodiinae, Dynastinae, Rutelinae, and Sericinae species with fewer available mitochondrial genome sequences in future studies to establish a more comprehensive phylogenetic relationship within Scarabaeidae, facilitating a better understanding of the topology, monophyly, polyphyly, and relationships among the subfamilies.

## 5. Conclusions

The complete mitogenome of *C. crataceus*, with a size of 17,946 bp, represents a typical circular, double-stranded DNA molecule, sharing key features with other reported Melolonthinae mitogenomes, such as gene order and tRNA secondary structures. The phylogenetic analyses provided strong support for the paraphyly of Melolonthinae and the monophyly of other Scarabaeidae subfamilies.

## Figures and Tables

**Figure 1 genes-16-00111-f001:**
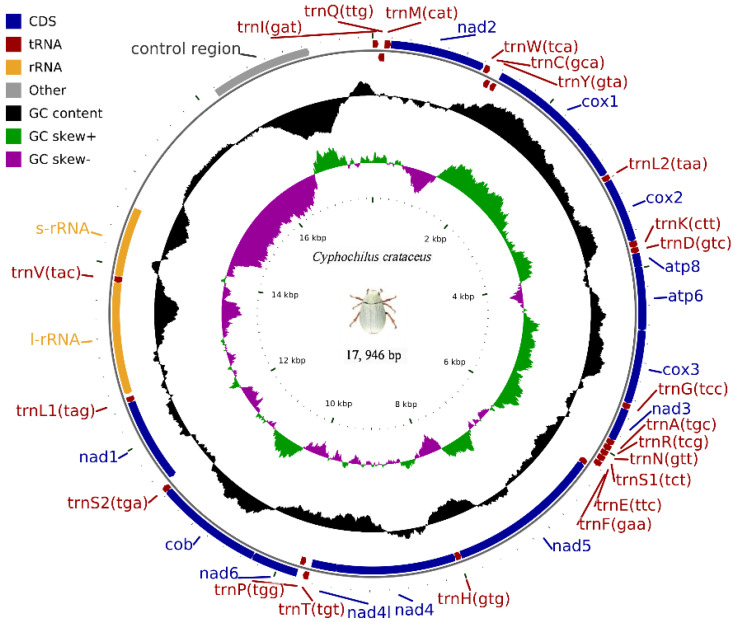
Mitochondrial genome structure of *C. crataceus*.

**Figure 2 genes-16-00111-f002:**
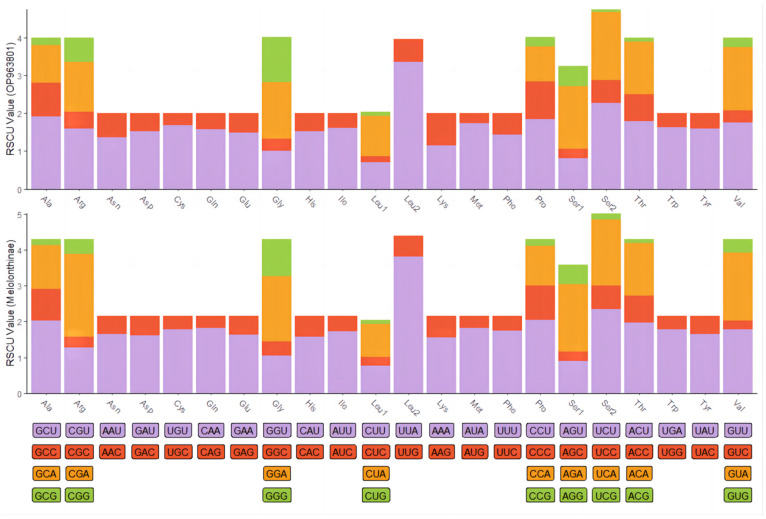
Relative synonymous codon usage (RSCU) in mitochondrial protein-coding genes of *C. crataceus* and 19 Melolonthinae species.

**Figure 3 genes-16-00111-f003:**
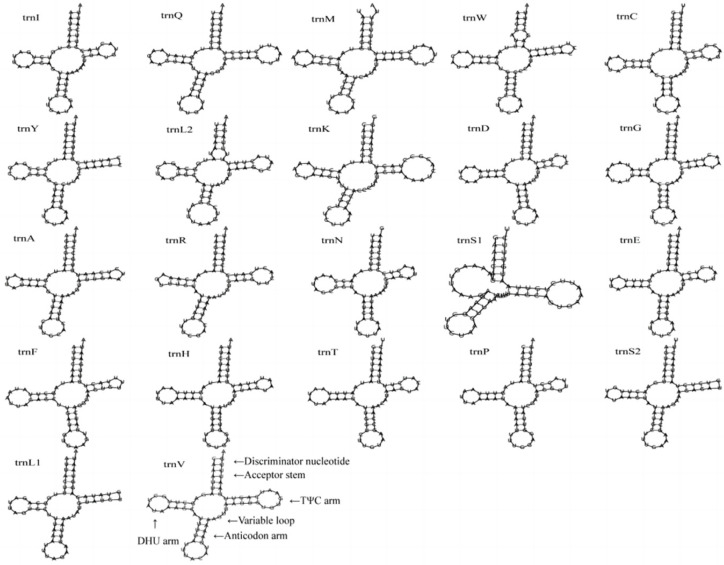
The secondary structures of *tRNA* genes inferred for *C. crataceus*.

**Figure 4 genes-16-00111-f004:**
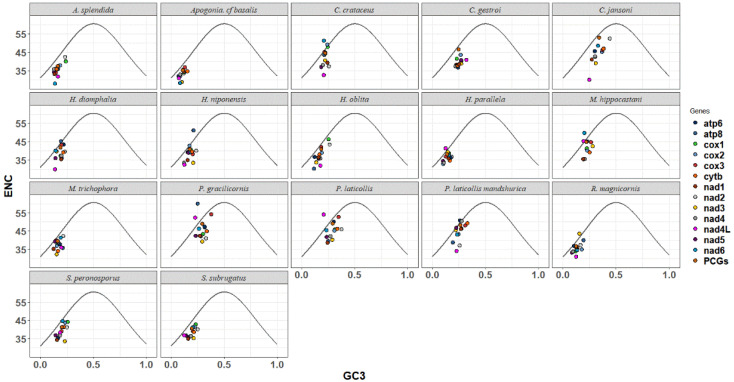
ENC-GC3 analysis of mitochondrial genomes of 17 species of Melolonthinae.

**Figure 5 genes-16-00111-f005:**
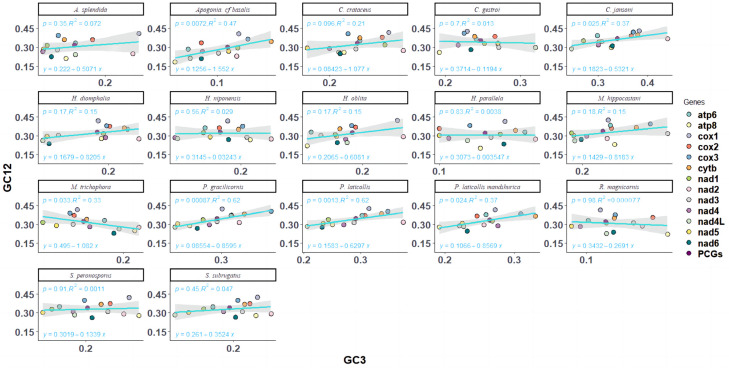
Neutrality plot analysis of mitochondrial genomes of 17 species of Melolonthinae.

**Figure 6 genes-16-00111-f006:**
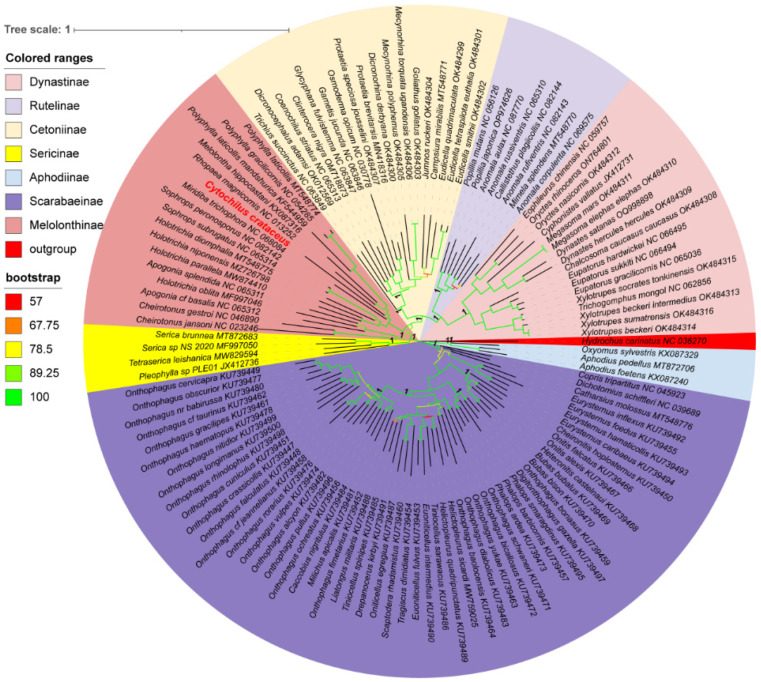
Phylogenetic tree of Scarabaeidae based on protein-coding genes and constructed by Bayesian inference and Maximum Likelihood methods(The species highlighted in red font is sequenced in this study).

**Table 1 genes-16-00111-t001:** Annotation of the complete mitogenome of *C. crataceus*.

Gene	Coding Strand	Position (bp)	Length (bp)	Start Codon	Stop Codon	Anticodon	Intergenic Nucleotide
*trnI*	N	1–63	63			GAT	−3
*trnQ*	J	61–129	69			TTG	−2
*trnM*	N	128–196	69			CAT	
*nad2*	N	197–1204	1008	ATT	TAG		14
*trnW*	N	1219–1284	66			TCA	−8
*trnC*	J	1277–1339	63			GCA	15
*trnY*	J	1355–1417	63			GTA	−8
*cox1*	N	1410–2951	1542	ATT	TAA		6
*trnL2*	N	2958–3021	64			TAA	
*cox2*	N	3022–3709	688	ATG	TAA		
*trnK*	N	3710–3780	71			CTT	
*trnD*	N	3781–3843	63			GTC	
*atp8*	N	3844–3999	156	ATC	TAA		−7
*atp6*	N	3993–4665	673	ATG	TAG		
*cox3*	N	4666–5452	787	ATG	TA(A)		
*trnG*	N	5453–5517	65			TCC	
*nad3*	N	5518–5871	354	ATA	TAG		−2
*trnA*	N	5870–5933	64			TGC	−1
*trnR*	N	5933–5997	65			TCG	
*trnN*	N	5998–6060	63			GTT	
*trnS1*	N	6061–6127	67			TCT	1
*trnE*	N	6129–6190	62			TTC	−2
*trnF*	J	6189–6253	65			GAA	−1
*nad5*	J	6253–7971	1719	ATT	TAA		
*trnH*	J	7972–8036	65			GTG	−1
*nad4*	J	8036–9370	1335	ATG	TAA		−7
*nad4l*	J	9364–9654	291	ATG	TAA		2
*trnT*	N	9657–9720	64			TGT	
*trnP*	J	9721–9782	62			TGG	10
*nad6*	N	9784–10,287	504	ATA	TAA		−1
*Cob* (*cytb*)	N	10,287–11,429	1143	ATG	TAG		−2
*trnS2*	N	11,428–11,490	63			TGA	17
*nad1*	J	11,508–12,458	951	ATG	TAA		1
*trnL1*	J	12,460–12,522	63			TAG	33
*rrnL*	J	12,556–13,807	1252				−3
*trnV*	J	13,805–13,874	70			TAC	−1
*rrnS*	J	13,874–14,657	784				
*OH*	N	14,658–17,946	3289				

Note: J: major strand; N: minor strand; intergenic nucleotide indicates the length of the intergenic sequence between this gene and the previous gene.

**Table 2 genes-16-00111-t002:** Base composition in the mitochondrial genome of *C. crataceus*.

Gene	A%	C%	G%	T%	A + T%	G + C%	AT Skew	GC Skew
Whole genome	37.64	18.30	9.88	34.19	71.82	28.18	0.048	−0.299
*nad2*	34.72	18.95	7.94	38.39	73.12	26.88	−0.050	−0.410
*cox1*	28.34	19.71	16.34	35.60	63.94	36.06	−0.114	−0.094
*cox2*	32.12	20.06	13.23	34.59	66.71	33.29	−0.0037	−0.205
*atp8*	37.82	16.67	7.69	37.82	75.64	24.36	0.000	−0.368
*atp6*	32.10	20.35	10.55	37.00	69.10	30.91	−0.071	−0.317
*cox3*	28.72	19.44	14.87	36.97	65.69	34.31	−0.126	−0.133
*nad3*	33.05	16.67	9.89	40.40	73.45	26.55	−0.100	−0.255
*nad5*	29.61	9.25	16.52	44.62	74.23	25.77	−0.202	0.282
*nad4*	28.24	9.29	16.78	45.69	73.93	26.07	−0.236	0.287
*nad4l*	24.74	8.25	16.15	50.86	75.60	24.40	−0.345	0.324
*nad6*	36.31	16.67	6.94	40.08	76.39	23.61	−0.049	−0.412
*cob*	31.06	19.95	12.51	36.48	67.54	32.46	−0.080	−0.229
*nad1*	24.89	9.63	20.35	45.13	70.02	29.98	−0.289	0.357
*rrnL*	35.70	7.59	16.37	40.34	76.04	23.96	−0.061	0.367
*rrnS*	36.10	8.93	17.86	37.12	73.21	26.79	−0.014	0.333

## Data Availability

The mitogenome sequences of *C. crataceus* have been deposited in GenBank under accession number OP963801.1.

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
