# Peer review of "Phylogenetic and Codon Usage Bias Analysis Based on mt-DNA of Cyphochilus crataceus (Coleoptera: Melolonthinae) and Its Neighboring Species"

_genes, 2025, doi:10.3390/genes16020111_

Round 1
Reviewer 1 Report
Comments and Suggestions for Authors
The manuscript entitled "Phylogenetic and Codon Usage Bias Analysis Based on mt-DNA of Cyphochilus crataceus (Coleoptera: Scarabaeidae) and Its Neighboring Species" provides a mitogenome sequence analysis of Cyphochilus crataceus. Additionally, the authors incorporated 121 other mitogenome sequences for a phylogenetic analysis of the Scarabaeidae family. Previous research has used both nuclear and mitochondrial genomes for phylogenetic analysis; however, the limitation of this study is that it relies solely on the mitochondrial genome to construct the phylogenetic relationships within Scarabaeidae. Although the manuscript is well-written, I was unable to assess any of the figures due to their lack of quality. Therefore, I would need to review the manuscript again to evaluate the illustrations.
Author Response
Comments 1: [The manuscript entitled "Phylogenetic and Codon Usage Bias Analysis Based on mt-DNA of Cyphochilus crataceus (Coleoptera: Scarabaeidae) and Its Neighboring Species" provides a mitogenome sequence analysis of Cyphochilus crataceus. Additionally, the authors incorporated 121 other mitogenome sequences for a phylogenetic analysis of the Scarabaeidae family. Previous research has used both nuclear and mitochondrial genomes for phylogenetic analysis; however, the limitation of this study is that it relies solely on the mitochondrial genome to construct the phylogenetic relationships within Scarabaeidae. Although the manuscript is well-written, I was unable to assess any of the figures due to their lack of quality. Therefore, I would need to review the manuscript again to evaluate the illustrations.]
Response 1: Thank you for pointing this out. We agree with this comment. Therefore, we have have submitted the images separately, and We hope you can view them. If you are unable to see the separately submitted images, you might also try enlarging the images in the Word document; they may appear clearer that way.

Reviewer 2 Report
Comments and Suggestions for Authors
Dear Authors,
The present article "Phylogenetic and codon usage bias analysis based on mt-DNA of Cyphochilus cretaceous (Coleoptera:Scarabaeidae) and its neighbouring species" is methodologically and graphically well presented. The main objective was to analyse the mt-DNA of one species of Cyphochilus cretaceous. As a result, the comparison study with other mtDNA of 121 species of Scarabaeidae of sequences downloaded from GeneBank did not add any real new significant scientific data. The results obtained confirm the previous data mentioned in the last part of the discussion.
Therefore, this study should be extended to other species of the mentioned subfamilies whose phylogenetic structures are unclear.
The authors underline that "the importance of sampling more Scarabaeidae species, especially the Aphodiinae, Dynastinae, Rutelinae, and Sericinae species with fewer available mitochondrial genome sequences" Such studies would be more significant at present, then suggestion of future studies.
The conclusion of this paper should be that this study brings the mitochondrial genome of Cyphochilus cretaceous similar to other species showed typical features of Scarabaeidae mitogenomes, what is obvious in most species of the family.
My main suggestion is to extend this study and confirm or not the family Melolonthidae. It could be really interesting and significant result.
Author Response
1. Summary |
|
|
Thank you very much for taking the time to review this manuscript. Please find the detailed responses below and the corresponding revisions/corrections highlighted/in track changes in the re-submitted files.
|
||
2. Point-by-point response to Comments and Suggestions for Authors |
||
Comments 1: [This study should be extended to other species of the mentioned subfamilies whose phylogenetic structures are unclear. The authors underline that "the importance of sampling more Scarabaeidae species, especially the Aphodiinae, Dynastinae, Rutelinae, and Sericinae species with fewer available mitochondrial genome sequences" Such studies would be more significant at present, then suggestion of future studies.]
|
||
Response 1: Thank you for pointing this out. We agree with this comment. Currently, most studies, including this one, support the phylogenetic relationship of Scarabaeidae as ((Scarabaeinae+Aphodiinae)+((((Dynastinae+Rutelinae)+Cetoniinae)+Melolonthinae)+Sericinae))+outgroup. The current findings indicate that, except for Melolonthinae, which is paraphyletic, the remaining groups are monophyletic. However, the existing data for the four groups—Aphodiinae, Dynastinae, Rutelinae, and Sericinae—is limited, and it is possible that with the addition of species, these groups may be re-evaluated as non-monophyletic. Therefore, we have made this suggestion.
|
||
Comments 2: [The conclusion of this paper should be that this study brings the mitochondrial genome of Cyphochilus cretaceous similar to other species showed typical features of Scarabaeidae mitogenomes, what is obvious in most species of the family. My main suggestion is to extend this study and confirm or not the family Melolonthidae. It could be really interesting and significant result.] |
||
Response 2: Thank you very much for your suggestions. Our initial motivation for studying the species Cyphochilus cretaceous was to explore the phylogenetic position of this relatively neglected species. Additionally, we investigated the codon usage of C. cretaceous and its family Melolonthidae, along with the factors influencing it. Our study found that various factors affect their codon usage; however, the codon usage of C. cretaceous is primarily influenced by genetic mutations. Furthermore, the codon usage preferences of different species within Melolonthidae are inconsistent, which indirectly reflects the paraphyly of this family.
|
Round 2
Reviewer 2 Report
Comments and Suggestions for Authors
The authors responded to the suggestions and improved the abstract and phylogenetic part of the discussion indicating that the codon usage of C. cretaceous is primarily influenced by genetic mutations.
However, the response to the suggestion [2] is unclear, and I thought it was a mistake, but they still meant the subfamily Melolonthinae.
Author Response
Comments 1: [The response to the suggestion [2] is unclear, and I thought it was a mistake, but they still meant the subfamily Melolonthinae. ]
|
Response 1: We sincerely apologize for the misunderstanding regarding your previous suggestion. I initially thought you were asking whether there is a polyphyly issue with Melolonthinae, but now I realize you were inquiring whether Cyphochilus crataceus belongs to Melolonthinae. According to the literature, Sabatinelli (2020a) was the first to propose, based on morphological characteristics, that Cyphochilus be classified within the subfamily Melolonthinae incerta sedis, and he provided morphological descriptions of several species within this genus, all of which were considered to belong to Melolonthinae. This classification has also been supported by Prakash (2023). Furthermore, based on molecular sequences uploaded to NCBI (Accession: OR753700.1; OR753671.1; OR753670.1; HG810087.1), the uploaders have identified Cyphochilus as belonging to Melolonthinae. Additionally, based on our phylogenetic tree, this study concludes that C. crataceus indeed belongs to Melolonthinae. Sabatinelli G. Taxonomic notes on the genus Cyphochilus Waterhouse, 1867 (Coleoptera, Scarabaeoidea, Melolonthinae) with description of 10 new species[J]. Revue suisse de Zoologie, 2020a, 127(1): 157-181 Sabatinelli G. Taxonomic notes on the genus Cyphochilus (Coleoptera: Scarabaeoidea: Melolonthinae)(part 3) with description of three new species from Indochina[J]. Acta Soc. Zool. Bohem, 2020, 84: 51-65. Sabatinelli G. Taxonomic notes on the genus Cyphochilus Waterhouse 1867 (Coleoptera, Scarabaeoidea, Melolonthinae)(part 2) with description of nine new species and a new subspecies[J]. Munis Entomology & Zoology, 2020, 15(2): 301-318. Sabatinelli G, Phạm P. Taxonomic notes on the genus Cyphochilus (Coleoptera: Scarabaeoidea: Melolonthinae)(part 4) with description of eight new species from Indochina and China[J]. Revue suisse de Zoologie, 2021, 128(1): 157-172. Prakash K V, Yeshwanth H M. Redescription of Cyphochilus niveosquamosus (Blanchard, 1851)(Coleoptera, Scarabaeidae, Melolonthinae) with notes on two species of the genus from India[J]. Entomon, 2023, 48(4): 567-574. |